# Fine-Tuning Language Models Using Formal Methods Feedback

**Yunhao Yang**[*1], **Neel P. Bhatt**[*1], **Tyler Ingebrand**[*1], **William Ward**[1], **Steven Carr**[1], **Zhangyang Wang**[1], **Ufuk Topcu**[1]

[1]The University of Texas at Austin, USA
{yunhaoyang234, npbhatt, tyleringebrand, stevencarr, atlaswang, utopcu}@utexas.edu

## Abstract

Although pre-trained language models encode generic knowledge that is beneficial for planning and control, they may fail to generate appropriate control policies for domain-specific tasks. Existing fine-tuning methods use human feedback to address this limitation. However, sourcing human feedback is labor-intensive and costly. We present a fully automated approach to fine-tune pre-trained language models for domain-specific applications, bridging the gap between generic knowledge and domain-specific requirements while reducing cost. The method synthesizes automaton-based controllers from pre-trained models guided by natural language task descriptions. These controllers are verifiable against independently provided specifications within a world model, which can be abstract or obtained from a high-fidelity simulator. Controllers with high compliance with the desired specifications receive higher ranks, guiding the iterative fine-tuning process. We provide quantitative evidence, primarily in autonomous driving, to demonstrate the method's effectiveness across multiple tasks. The results indicate an improvement in the percentage of specifications satisfied by the controller from 60% to 90%.

## Introduction

Pre-trained language models encode rich world knowledge that is useful for planning and control. Recent works use pre-trained models to synthesize control policies for tasks such as autonomous driving (Seff et al. 2023), surgical robotics (Janssen, Kazemier, and Besselink 2023), and aircraft operation (Tikayat Ray et al. 2023). The control policies yield high-level actions that an agent should take in order to satisfy objectives specified via natural language prompts.

However, in specific domains, pre-trained models may fail to generate appropriate control policies. For instance, an autonomous driving system may require knowledge about traffic rules and conventions specific to a given country. Such specific rules and conventions may be beyond the knowledge encoded in the pre-trained model.

To address this shortcoming, several works use human feedback for fine-tuning pre-trained models and to incorporate required domain knowledge (Stiennon et al. 2020; Christiano et al. 2017; Rafailov et al. 2023). Human feedback evaluates the extent to which the output of a pre-trained model aligns with the desired objectives. For example, the provision

---

[*]These authors contributed equally.

of a binary ranking of like or dislike for each model output can act as a feedback source. This feedback from human expertise enables fine-tuning of the pre-trained model and allows implicit incorporation of domain-specific knowledge. However, obtaining feedback from humans is labor-intensive and costly.

We investigate how similar feedback can be automatically obtained using artifacts from formal methods. Suppose we have a world model, which is either abstract or obtained from a high-fidelity simulator, and a set of specifications. We can verify, either formally or empirically, if a controller generated by the language model meets the specifications (Yang et al. 2022). The measure of compliance can act as a source of feedback for fine-tuning, similar to human feedback. Since this procedure is automated, such feedback is less labor-intensive and cheaper.

We develop a method to fine-tune pre-trained models based on automated feedback using artifacts from formal methods. The proposed method synthesizes an automaton-based controller from the pre-trained model given a natural language task description (Yang et al. 2022). Such an automaton-based controller is formally verifiable against independently provided specifications (e.g., a driving rule book (Censi et al. 2019)) when implemented in a specific world model. We can obtain the number of specifications satisfied by each controller and use it for ranking. We then iteratively fine-tune the pre-trained model using this ranking as a feedback source.

If the world model is obtained from a high-fidelity simulator rather than an abstract model, we collect trajectories from the simulator. The trajectories are sequences of state-action pairs which can be checked against the provided specifications. A controller satisfying a larger number of specifications when executed in the simulator is assigned a higher rank. We use the obtained ranks for fine-tuning.

To demonstrate the performance of the proposed method, we provide experimental results covering multiple tasks in an autonomous driving system, although applicability is not limited to this domain. The quantitative results indicate a significant improvement in the percentage of specifications satisfied, from 60% to above 90%, confirming that the proposed method can effectively fine-tune the pre-trained model.

## Related Work

**Fine-tuning from Human Feedback.** Reinforcement learning from human feedback (RLHF) is a preference alignment strategy that learns a reward model from human preferences and then fine-tunes the pre-trained language model using reinforcement learning (Stiennon et al. 2020). In some works, before fine-tuning begins, humans compare the accuracy of multiple responses to a single input and indicate which is preferred, generating a data set of preferences that is used to train a reward function (Stiennon et al. 2020; Ouyang et al. 2022). Other methods optimize the reward function and fine-tune the language model simultaneously. As the model generates outputs, a human indicates which output is preferred, sending new feedback for the reward function to learn, thus impacting the model's accuracy (Christiano et al. 2017).

Direct preference optimization (DPO) is a preference alignment strategy that implicitly optimizes the same objective as RLHF without explicitly learning a reward model or using reinforcement learning. DPO optimizes model outputs directly from human feedback data using a modified maximum likelihood objective, reducing the number of training stages and improving stability (Rafailov et al. 2023).

However, all of the above works rely on humans to provide feedback on which outputs are preferred. Obtaining an excessive amount of human feedback is labor intensive. In contrast, the method we propose automatically ranks the outputs from language models. Hence we can obtain an unlimited number of data points to fine-tune the language model.

**Fine-tuning from Generated Outputs** Some methods fine-tune a language model using the outputs of another model. For example, a language model can learn how to generate common sense phrases (Zhou, Bras, and Choi 2023) or output chain-of-thought reasoning (Li et al. 2023) using responses from a model that already exhibits the desired behavior. Other methods train a language model using the model's own outputs by identifying high-quality statements and feeding them back into the model as examples of correct responses (Bhagavatula et al. 2023; Jung et al. 2023). One approach combines both methods, first fine-tuning using the outputs of a separate pre-trained model, and then fine-tuning again using the model's own filtered outputs (Jung et al. 2023). Another strategy is to modify the backpropagation process so that only certain parameters are updated (Chen et al. 2020).

These methods are not capable of fine-tuning domain-specific language models since all the generated outputs from itself or other models lack domain-specific knowledge as well. In contrast, the method we proposed can fine-tune the language model to satisfy domain-specific requirements.

**Formal Methods and Verification on Language Models.** Existing works convert natural language to formal language, which can be used for verification (Baral et al. 2011; Sadoun et al. 2013; Ghosh et al. 2016). Recent works show that language models can be trained to convert natural language to formal language, with applications in representing mathematics, generating proofs, and creating assurance cases (Hahn et al. 2022; Wu et al. 2022; First et al. 2023; Chen, Deng, and Du 2023). One method is to design input prompts that include task-specific information (e.g., definitions, response

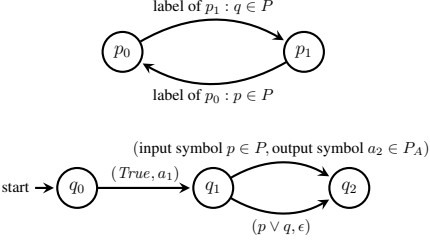

Figure 1: Examples of an automaton-based model (top) and a controller (bottom).

templates, and detailed examples) that enable a language model to divide a goal into individual steps and develop formal constraints on the system (Chen, Deng, and Du 2023). Other methods iteratively refine the input prompt to the language model based on counter-examples until the outputs pass formal verification (Jha et al. 2023; Yang et al. 2022). Although these works utilize formal methods, there are still humans in the loop, while our proposed method aims to be fully automated without any human intervention.

## Preliminaries

**Automaton-Based Model for System or Environment.** A model is an abstract representation that encodes the static and dynamic information of a system or an environment. We use a transition system to build the model.

A transition system $\mathcal{M} := \langle \Gamma_{\mathcal{M}}, Q_{\mathcal{M}}, \delta_{\mathcal{M}}, \lambda_{\mathcal{M}} \rangle$ consists of a set of output symbols $\Gamma_{\mathcal{M}}$, a set of states $Q_{\mathcal{M}}$, a non-deterministic transition function $Q_{\mathcal{M}} \times Q_{\mathcal{M}} \to \{0, 1\}$, and a labeling function $\lambda_{\mathcal{M}} : Q_{\mathcal{M}} \to \Gamma_{\mathcal{M}}$.

We introduce a set of atomic proposition $P$ such that $\Gamma_{\mathcal{M}} := 2^P$, i.e., a symbol $\sigma \in \Gamma_{\mathcal{M}}$ is the set of atomic propositions in $P$ that evaluate to *True*. Each symbol $\sigma$ captures the system or environment behavior. We present an example in Figure 1. We will leverage the fact that automaton-based structures are formally verifiable in the proposed method.

**Automaton-Based Controller.** A controller is a system component responsible for making decisions and taking actions based on the system's state. A controller can be mathematically represented as a mapping from the system's current state to an action, which is executable in the task environment. We use a *finite state automaton* (FSA) to build a controller for a sequential decision-making task.

A FSA is a tuple $\mathcal{A} = \langle \Sigma, A, Q, q_0, \delta \rangle$ where $\Sigma$ and $A$ are the sets of input and output symbols, $q_0 \in Q$ is the initial state, and $\delta : Q \times \Sigma \times A \times Q \to \{0, 1\}$ is a non-deterministic transition function. The transition function is a membership function—a transition exists when it evaluates to 1.

Each input symbol $\sigma \in \Sigma$ is composed of the atomic propositions from $P$, which is the set of atomic propositions we introduced for the model. We introduce another set of atomic propositions $P_A$ for the output alphabets $A := 2^{P_A}$. We also allow for a "no operation/empty" symbol $\epsilon \in A$. Note that the input symbols comprise all possible dynamics of the environment or system in which the controller operates,

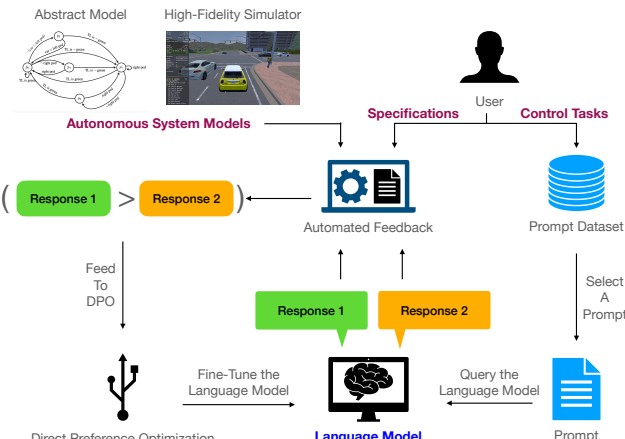

Figure 2: The overall pipeline of fine-tuning a language model via automated feedback. We mark the inputs to the pipeline in purple and the output in blue.

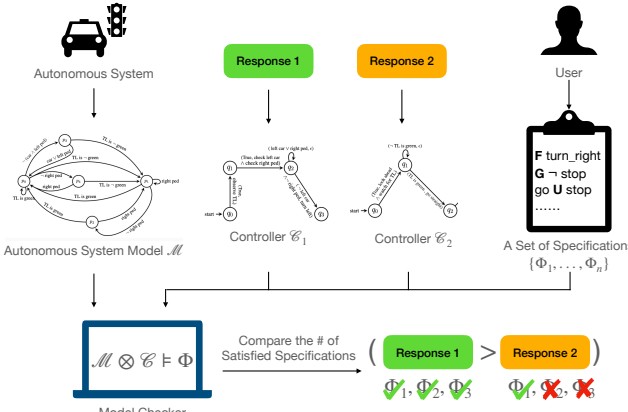

Figure 3: This diagram depicts the method of ranking responses by formal verification of the induced automata. We present the sample automata in Figures 5 and 7.

and the output symbols comprise all the actions allowed by the controller. See Figure 1 for an example.

## Fine-Tuning Pre-trained Language Models for Autonomous Systems

We develop a method for fine-tuning pre-trained language models for specific control tasks, named *direct preference optimization via automated feedback* (**DPO-AF**). The method first obtains human-provided information regarding the autonomous system. It then constructs a model that encodes the information about the system. Next, we query the pre-trained language model on a particular system control task and get multiple responses from the language model via sampling. We construct automata from the responses and apply verification methods to check how many user-provided specifications each automaton satisfies. We rank the responses by the number of satisfied specifications. Last, we send the prompt and ranked responses to the DPO algorithm to fine-tune the language model.

DPO-AF does not require repeated feedback from humans. Therefore, we can obtain an unlimited number of prompt-response pairs until the language model converges.

## Automaton-Based Representation for Natural Language and Autonomous Systems

**Modeling the Autonomous System.** DPO-AF starts by constructing an automaton-based model encoding the information about the autonomous system. Such information is obtained from external sources such as human experts or system operation manuals. The information includes but is not limited to a set of propositions that describe the system's behaviors and a set of control signals (actions) that can affect the system's states. We encode the set of behaviors in an atomic proposition set $P$ and the set of actions in an atomic proposition set $P_A$.

Recall that a model consists of a set of states, a set of symbols, a transition function, and a label function. As we

defined $P$ and $P_A$, we build $2^{|P|}$ states whose label is $\sigma \in 2^P$ respectively. $|P|$ is the number of propositions in $P$. Next, for every two states $p_i$ and $p_j$, we check whether the system supports the transition between the label of $p_i$ and the label of $p_j$. If the system supports such a transition, we add it into the transition function.

Finally, we remove the states with no incoming and outgoing transitions. However, from a conservative perspective, we can build transitions for every pair of states and not remove any states. The conservative approach can avoid potential missing transitions but will significantly increase the computation cost for formal verification.

To illustrate the procedure, suppose there is a traffic light system operating in the order of red-green-yellow-red. We have the proposition set $P = \{green, yellow, red\}$ and transitions (green to red), (red to yellow), and (yellow to green). Hence, we only keep three states with labels $green, yellow, red$ respectively and remove all the states with other labels (e.g., $green \wedge yellow$).

**Task Prompt Engineering.** Prior to fine-tuning the language model, we collect a prompt dataset. The prompt dataset consists of the queries on the control tasks that operate in the autonomous system.

Then, we define a prompt engineering procedure to extract relative task knowledge from the language model. For each prompt in the prompt dataset, we first use the following format to obtain the responses from the language model:

```
1  Define the steps for task description
2  1. step one description
3  2. step two description
4  ...
```

Blue texts and red texts indicate the input prompt and the language model's responses, respectively. This format forces the outputs to be a list of step descriptions for the task described in the input prompt.

Once we get the responses, we query the language model again to map the step descriptions into the set of defined

atomic propositions $P$:

```
1  Align the following steps to align the set
       of Boolean propositions {prop 1,...,
       prop n} and actions {act 1, ..., act m}:
2  1. step one description
3  2. step two description
4  ...
5
6  1. aligned step one description
7  ...
```

We rephrase the step description so that the propositions and actions are consistent with the model. Therefore, we avoid failing the verification process due to language ambiguity, i.e., different phrases with the same meaning.

Note that DPO-AF also aims to fine-tune the language model to output steps that can be easily aligned to the propositions and actions. Therefore, the expected responses from the fine-tuned language model should have the following properties: 1. The language model can easily and correctly align the textual step descriptions to the given propositions and actions. 2. The aligned step descriptions satisfy the user-provided specifications.

To check the second property, we need to construct an automaton-based controller from the textual step descriptions. Then, we implement the controller in the model and verify it against the specifications.

**Controller Construction.** We follow the method GLM2FSA (Yang et al. 2022) to construct an FSA-based controller to encode the textual step descriptions. Specifically, we start from the aligned textual step descriptions and apply semantic parsing to break the steps into a list of verb phrases. Recall that a controller consists of a set of states $Q$, an initial state $q_0$, input symbols $\Sigma$, output symbols $A$, and a transition function $\delta$. We use the verb phrases to define the input and output symbols according to the grammar from GLM2FSA. Then, we build one state corresponding to each step, with the state corresponding to the first step as the initial state. Last, we follow the GLM2FSA algorithm to build the transition rules. We present a step-by-step illustrative example in Section .

## Automated Feedback

Given a set of specifications, we provide two ways of checking whether the controllers constructed from the language model's outputs satisfy each specification. For each output, the method generates feedback consisting of the number or percentage of specifications being satisfied.

**Formal Verification.** Formal verification requires an automaton-based model, an automaton-based controller, and a set of logical specifications. So far, we have constructed the model and the controller. The specifications include the expectation of task achievement or safety requirements, represented in temporal logic (e.g., linear temporal logic (Pnueli 1977)). The temporal logic specifications are logic formulas over propositions $P \cup P_A$. We describe it in detail in the Appendix. These specifications are either provided by the task designer or extracted from existing rule books.

In the verification procedure, we first implement the controller in the model. Mathematically, we define a product automaton $\mathfrak{P} = \mathcal{M} \otimes \mathcal{C}$ describing the interactions of the controller $\mathcal{C}$ with the model $\mathcal{M}$, i.e., how the controller's actions change the model's states and how the model's states affect the controller's decision-making on its next action. Note that the verification procedure implicitly assumes that all the actions can be successfully operated and hence lead to the corresponding states of the controller and the model.

We run a model checker (e.g., NuSMV (Cimatti et al. 2002)) to verify if the product automaton satisfies each specification,

$$\mathcal{M} \otimes \mathcal{C} \models \Phi. \qquad (1)$$

We verify the product automaton against each specification for all the possible initial states. If the verification fails, the model checker returns a counter-example. The counter-example is a trace—a sequence of states—violates the specifications. The NuSMV model checker returns the sequence of states from the product automaton along with the output symbols. Mathematically, the traces are in a format of $(p_1, q_1, c_2 \cup a_1), (p_2, q_2, c_2 \cup a_2), ...$ where $p_i \in Q_{\mathcal{M}}, q_i \in Q, c_i = \lambda_{\mathcal{M}}(p_i), a_i \in A$ such that $\delta(q_i, \lambda_{\mathcal{M}}(p_i), a_i, q_{i+1}) = 1$.

We present the definitions of temporal logic and product automaton in the Appendix.

**Empirical Evaluation.** In some scenarios, obtaining models for autonomous systems may be hard. We propose using empirical evaluation to account for the scenarios where models are not present. Empirical evaluation requires an autonomous system $\mathcal{S}$, a constructed controller $\mathcal{C}$, an atomic proposition set $P$, a set of actions $P_A$, and a grounding method $\mathbf{G}$. Specifically, $\mathbf{G} : \mathcal{C} \times \mathcal{S} \to (2^P \times 2^{P_A})^N$ operates the controller directly in the system and returns a sequence of propositions and actions describing the operation. $N$ is the max length of the sequence. The sequence is evaluated as follows:

$$\mathbf{G}(\mathcal{C}, \mathcal{S}) = (2^P \times 2^{P_A})^N \models \Phi. \qquad (2)$$

After evaluating every sequence against the specifications, we get the percentage of sequences, $\mathbb{P}_\Phi$, which satisfy each specification:

$$\mathbb{P}_\Phi = \frac{\text{number of sequences satisfying } \Phi}{\text{total number of sequences}}.$$

## Fine-Tuning via Automated Feedback

**Collection of the Language Model's Outputs.** Once we select the autonomous system and obtain the model for the system, we can query the model for instructions on tasks that are operable in the system, following the format described in the previous section. Different responses for the same input task can be sampled from the language model. Then, we can rank these responses and fine-tune the language model to output the best response according to the system model.

**Ranking the Outputs and Fine-Tuning the Language Model.** We apply the automated feedback method for every two responses from the language model associated with the same task prompt to rank the preferences of the two responses. As a result, we obtain a data point $(x, y_w, y_l)$, where $x$ is the input prompt, $y_w$ is the preferred response and $y_l$ is

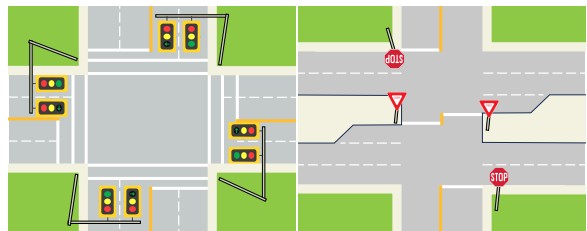

Figure 4: Illustration of two sample scenarios from the autonomous driving system. The left figure is an intersection with the traffic light. We encode this scenario in a model in Figure 5. The right figure is an intersection with a wide median. We encode it in a model in Figure 6.

the unpreferred response. For a given set of specifications, we construct a controller from each response and verify it against each of the specifications. The response satisfying more specifications (or having a higher percentage $\mathbb{P}_\Phi$ of sequences satisfying the specifications) is preferred. If we have collected $N$ tasks and $m$ responses per task. Then, we will have a maximum number of $N \times C_2(m)$ data points, where $C_i(j)$ means $j$ choose $i$.

Then, we feed the pairs of responses, along with their prompt, to the DPO algorithm. The DPO algorithm fine-tunes the parameters of the language model accordingly. During fine-tuning, we use low-rank approximation to reduce computational complexity (Hu et al. 2021).

## Experimental Results

To validate the proposed method, we apply DPO-AF on Llama2-7B for controlling an autonomous driving system. We first provide a demonstration of how we obtain the verification feedback. Then, we present quantitative results to show the effectiveness of DPO-AF at the mathematical level. Next, we use an autonomous driving simulator, Carla (Dosovitskiy et al. 2017), to show Llama's performance enhancement at the operation level. Lastly, we provide evidence that the generated controller can be transferred to real-life, indicating the applicability of our approach.

### Example Demonstration

**Examples of System Modeling.** To obtain formal verification feedback for the language model's outputs, we first construct automaton-based models that encode the information of the autonomous driving system. Such information includes the objects from the environment and potential environment dynamics that can be perceived by the autonomous vehicle. Note that the models are externally provided, either from human expertise or system manuals.

Figure 5 and 6 show the automaton-based models encoding the information on a regular traffic light intersection and a wide median (which we present in Figure 4). We construct one model for each scenario in the autonomous driving system. We integrate these models together to form a universal model representing the entire system. In this way, we can later implement the constructed controllers into the model for

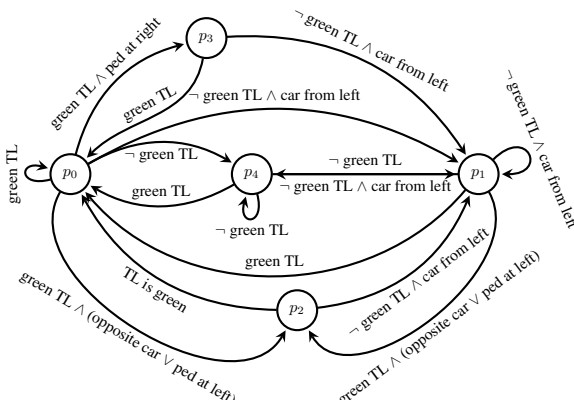

Figure 5: An automaton-based model represents a vehicle's environment dynamics at a regular traffic signal at an intersection. TL represents "traffic light," and ped represents "pedestrian."

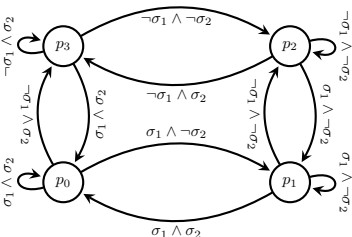

Figure 6: An automaton-based model representing the environment dynamics of a yield-based wide median. $\sigma_1 =$ car from left and $\sigma_2 =$ car from right.

formal verification. We present models for other scenarios in the autonomous driving system in the Appendix.

**Examples of Externally Provided Specifications.** For verification purposes, we generate a set of traffic rules in the form of temporal logic. We denote the traffic rules as *specifications*. Some examples from the set of specifications in the temporal logic formula are presented below:

$\Phi_1 = \Box(\text{pedestrian} \rightarrow (\Diamond \text{stop}))$,

$\Phi_2 = \Box(\neg\text{turn left} \lor (\neg\text{opposite car} \lor \text{green left-turn light})$,

$\Phi_3 = \Box(\neg\text{green traffic light} \rightarrow \neg\text{go straight})$,

$\Phi_4 = \Box(\text{stop sign} \rightarrow \Diamond \text{stop})$,

$\Phi_5 = \Box \neg\text{turn right} \lor \neg(\text{car from left} \lor \text{pedestrian at right})$,

We present the full set of specifications in the Appendix.

From the provided models and specifications, we can extract a set of atomic propositions and a set of actions. We add the English vocabulary from the model's input symbols to the set of atomic propositions. We add any vocabularies from the temporal logic formulas that are not already in the proposition set to the action set. Now, we have obtained a set of atomic propositions and allowable actions from the model and specifications. The propositions include { green traffic light, green left-turn light, flashing left-turn light, opposite

car, car from left, car from right, pedestrian at left, pedestrian at right, pedestrian in front, stop sign }, and the actions are { stop, turn left, turn right, go straight }.

Once we have the models representing the autonomous system and a set of specifications, we can construct controllers for various tasks that can operate in the system and verify whether the controllers satisfy the specifications when operating in the system.

**Example on Controller Construction: Right Turn.** In this example, we use a simple task "turn right at the traffic light" to demonstrate the controller construction and verification. Then, we compare the controllers constructed using the language model prior to and after the fine-tuning stage. Hence we can give a straightforward demonstration of how DPO-AF improves the language model.

We start by constructing a controller using the language model before fine-tuning it. First, we query the language model on the steps of turning right at the traffic light in the following format:

```
1  Steps for "turn right at traffic light"
2  1. Look straight ahead and watch for traffic
      light.
3  2. If the traffic light turns green, start
      moving forward.
4  3. As you approach the intersection, look to
      your left for oncoming traffic.
5  4. If there is no traffic coming from your
      left, check pedestrians on your right.
6  5. If it is safe, turn your vehicle right.
```

Next, we again query the language model to align the step instructions to the defined propositions and actions:

```
1  Rephrase the following steps to align the
      defined Boolean Propositions {green
      traffic light, car from left, ......}
      and Actions {stop, turn left, turn right
      , go straight}:
2  1. Look straight ahead and watch for traffic
      light.
3  ......
4  5. If it is safe, turn your vehicle right.
5
6  1. Observe the state of the green traffic
      light.
7  2. If the green traffic light is on, execute
      the action go straight.
8  3. As you approach the intersection, observe
      the state of the car from left.
9  4. If the car from left is not present,
      check the state of the pedestrian at
      right.
10 5. If the pedestrian at right is not present
      , execute the action turn right.
```

By doing so, we have aligned the vocabularies from the initial responses to the defined propositions and actions. Note that such alignment may also introduce mistakes that violate more specifications, but reducing the probability of an alignment mistake is also a goal for DPO-AF.

Then, we apply semantic parsing to break the sentence

into verb phrases and keywords (e.g., if) and then shorten the phrases for presentation purposes:

```
1  1. <observe traffic light>.
2  2. <if> <green traffic light>, <go straight>
      .
3  3. <observe car from left>.
4  4. <if> <no car from left>, <check
      pedestrian at right>.
5  5. <if> <no pedestrian at right>, <turn
      right>.
```

Last, we follow the algorithm GLM2FSA (Yang et al. 2022) to construct an FSA representing the steps of this task, as presented in the left of Figure 7.

```
1  Steps for "turn right at traffic light"
2  1. Observe the traffic light in front of you
      .
3  2. Check for the left approaching car and
      right side pedestrian.
4  3. If no car from the left is approaching
      and no pedestrian on the right, proceed
      to turn right.
```

**Example on Formal Verification** We first implement both controllers in the automaton-based model presented in Figure 5, i.e., construct a product automaton for each controller and the model.

Second, we verify both product automata against the set of provided specifications. During the verification step, the model checker finds that the controller obtained before fine-tuning fails the specification $\Phi_5$. The model checker returns a counter-example on states $(p_0, q_3), (p_4, q_4), (p_1, q_5)$.

This counter-example captures an edge case: The traffic light turns back to red and a car is coming from the left immediately after the agent is checking or waiting for pedestrians. In this scenario, the agent does not check for the traffic light and cars from left again and directly turns right, which can lead to an accident. We argue that this edge case can hardly be caught by human inspection but can be found by the model checker. Hence we highlight this counter-example to indicate the necessity of formal verification.

In contrast, the controller obtained after fine-tuning satisfies all the specifications. Through this right-turn example, we observe the language model's enhancement through DPO-AF. We present more controller construction and verification examples in the Appendix.

## Quantitative Evaluation

**Fine-tuning via DPO.** DPO fine-tunes a language model to output responses that match desired specifications. DPO requires a data set where each data point has the form $(x, y_w, y_l)$, where $x$ is a user input (i.e., "Steps for turn right at the traffic light"), $y_w$ and $y_l$ are the language model's text responses such that the user prefers $y_w$ over $y_l$. In our experiments, the preferred response is the one whose FSA-based representation satisfies more of the specifications than the other response. We collect approximately 3000 data points to fine-tune the language model. After fine-tuning, the language

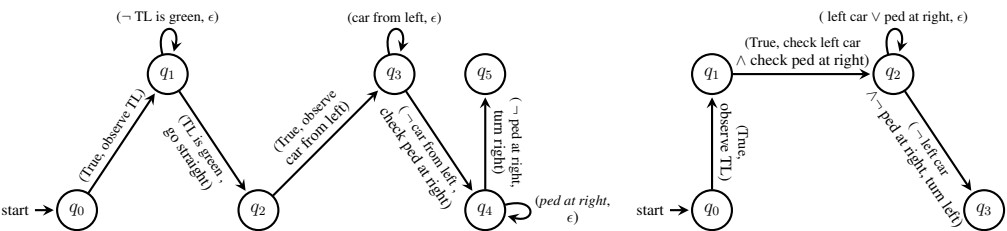

Figure 7: Automaton-based controllers for the task "turn right at the traffic light." The left controller is obtained before fine-tuning the language model, and the right controller is obtained after the fine-tuning. TL represents "traffic light."

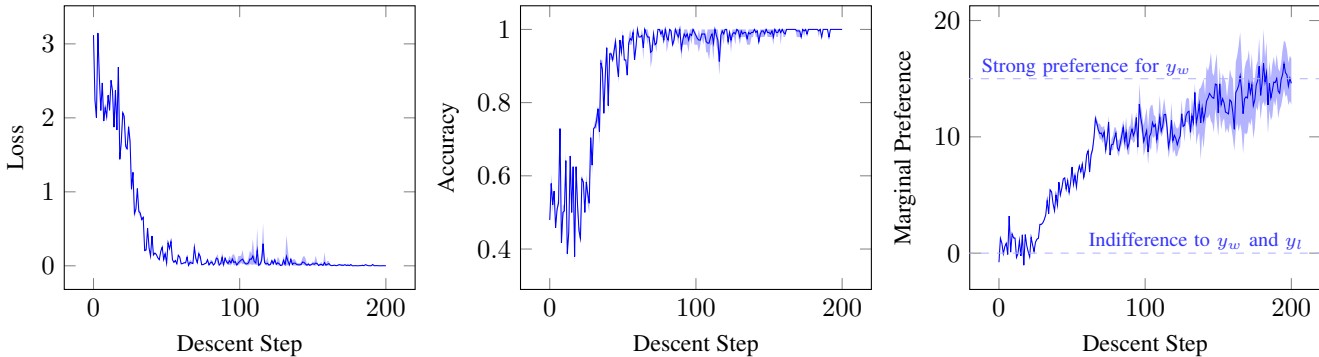

Figure 8: This figure shows fine-tuning statistics for Llama2-7B optimized for an autonomous driving system. All plots show the mean over five seeds. Shaded areas indicate maximum and minimum values. Plots from left to right show the DPO losses, accuracies, and marginal preferences over different epochs, respectively.

model shows a preference for the responses as indicated in the training dataset.

We measure the DPO training performance via three metrics: DPO loss, accuracy, and marginal preference. Loss refers to the modified maximum likelihood loss function from the DPO algorithm, which is minimized via gradient descent. Accuracy measures how often the model prefers the correct response over the incorrect response. Accuracy is the mean over the dataset of $\mathbb{I}(P(y_w|x,\theta) > P(y_l|x,\theta))$, where $\mathbb{I}$ is the indicator function returning one if the input is true and zero otherwise, and $\theta$ is the current values of the model parameters. Marginal preference measures how strongly the model prefers the correct output compared to the original reference model. Marginal preference is calculated as the mean over the dataset of $(log(P(y_w|x,\theta)) - log(P(y_w|x,\theta_{ref}))) - (log(P(y_l|x,\theta)) - log(P(y_l|x,\theta_{ref})))$. Zero indicates indifference, positive values indicate stronger preferences for the favored answer, and negative values indicate preference for the less preferred response.

We show the fine-tuning performance on the Llama2-7B model over the three metrics in Figure 8. Note that the variance between random seeds is relatively small because the model starts with the same parameters, and only the order of the data changes between seeds.

**Evaluation via Formal Verification.** We provide an additional metric to evaluate the proposed DPO-AF. During the fine-tuning procedure, we save a checkpoint language model for every 20 epochs. For each checkpoint language model,

we query it for various autonomous driving tasks and obtain the task controllers. Then, we verify the controllers against 15 provided specifications (presented in the Appendix) following the formal verification method in Section . Thus, we obtain the number of specifications being satisfied for each controller.

Figure 9 shows the relationship between the number of satisfied specifications and the number of epochs of DPO training. Simultaneously, we divide the results into two categories—training and validation—depending on whether the task is included in the training dataset. Hence, we have shown the relationships between the numbers of satisfied specifications and epochs for both training data and validation data.

For both training and validation data, we observe an increase in the number of specifications satisfied as we fine-tune for more epochs. This result indicates that our approach can improve the language model's ability to satisfy critical requirements. Therefore, our approach can act as a starting point to guide the design process for real-world implementations of autonomous driving systems.

**Justification for Overfitting.** We design the method DPO-AF to fine-tune language models for solving domain-specific tasks rather than enhancing the language model in general. Therefore, we do expect some degree of overfitting on the language model to the domain-specific knowledge and vocabulary. In our experiments, we fine-tune the language model specifically for tasks operated in autonomous driving systems.

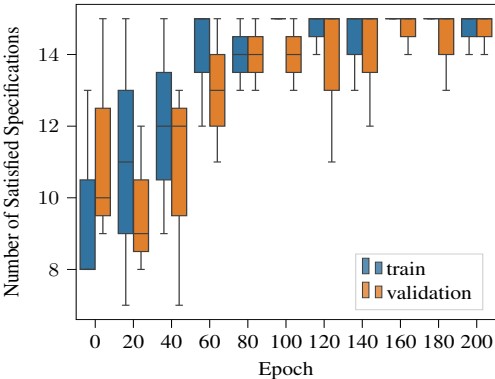

Figure 9: The number of specifications satisfied through formal verification vs. the epoch of DPO training.

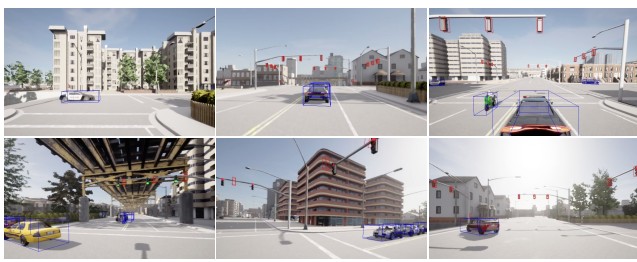

Figure 10: Visual demonstration of obtaining system information while operating the controllers. We use Carla to simulate the autonomous driving system.

A certain degree of overfitting provides stronger guarantees that the generated outcomes satisfy critical specifications.

**Empirical Evaluation in a Simulated System.** We have presented another approach to obtain feedback via empirical evaluation in Section . We will show consistency between feedback from empirical evaluation and formal verification.

As we obtain the controllers through the proposed method, we operate the controllers in the Carla simulator to collect operation data. Carla is a simulator for the autonomous driving system. During each operation of each controller, we obtain a sequence of propositions and actions—in the form of $(2^P \times 2^{P_A})^N$. The propositions come from the information returned by the autonomous system, and the actions come from the controller. The Carla simulator allows for the extraction of system information. We present visual demonstrations of extracting the propositions from the system in Figure 10. Then, we verify the sequence against the provided specifications, as we described in Section  under Empirical Evaluation. We operate the controllers multiple times in the system and verify the sequences against the specifications. For each specification, we get a percentage of the number of sequences satisfying this specification.

Figure 11 compares these percentages obtained before fine-tuning and after fine-tuning. Note that we run multiple controllers and collect multiple sequences for each controller. We show the results for the first five specifications as presented in Section .

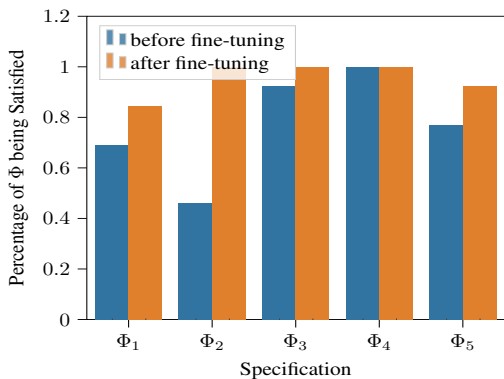

Figure 11: Percentage $\mathbb{P}_\Phi$ of each specification $\Phi$ being satisfied during actual operations in the system.

We observe that the percentages after fine-tuning are consistently higher than before fine-tuning among all five specifications, which means all the specifications have a higher probability of being satisfied for a given execution after fine-tuning. In Figure 9, we show that outputs from the fine-tuned model (at epoch 200) satisfy more specifications compared to the pre-trained model (at epoch 0). Hence, we obtain consistent feedback from the formal verification and empirical evaluation. Therefore, if we are unable to obtain automaton-based models for the system, empirical evaluation is a substitute for formal verification and is able to provide feedback consistent with formal verification.

From another perspective, this result provides additional evidence to show the effectiveness of the method DPO-AF, as it improves the probability of all the specifications being satisfied during operation.

## Conclusions

We develop a method of fine-tuning pre-trained language models via automated feedback for domain-specific tasks, such as control tasks in autonomous systems. The method converts the outputs from the pre-trained language model to automaton-based controllers. Then, it verifies how many of the externally provided specifications are satisfied by each controller. We rank the pre-trained language model's outputs by the number of satisfied specifications and feed these ranked outputs to the DPO algorithm for fine-tuning. We substitute human feedback with automated feedback using formal methods, which significantly decreases labor intensity. We provide empirical evidence on a simulated autonomous driving system to demonstrate the effectiveness of the proposed method: The fine-tuned language model satisfies more specifications compared with the model before fine-tuning.

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

# Additional Background and Definitions

**Temporal Logic.** Temporal logic is a formalism that describes properties of sequences over time, specifically in systems' behaviors. It employs logical operators that capture temporal aspects such as "always," "eventually," and "until" to specify constraints and requirements on system executions. Formally, temporal logic formulas are defined inductively as: $\varphi := p \in P_{\mathcal{M}} \mid \neg\varphi \mid \varphi \vee \varphi \mid \circ\varphi \mid \varphi \, \mathbf{U} \, \varphi$ Intuitively, a temporal logic formula consists of a set of atomic propositions, a set of temporal operators, and a set of logical connectives.

The common-used temporal operators are $\Diamond$ ("eventually"), $\mathbf{U}$ ("until"), $\circ$ ("next"), and $\square$ ("always"). And the logical connectives includes $\wedge$ ("and"), $\vee$ ("or"), $\neg$ ("not"), etc.

**Product Automaton.** Let a controller be $\mathcal{C} := \langle \Sigma, A, Q, q_0, \delta \rangle$ with input alphabet $\Sigma := 2^P$, output alphabet $A := 2^{P_A}$, and non-deterministic transition function $\delta : Q \times \Sigma \times A \times Q \to \{0, 1\}$.

Let a model be a tuple $\mathcal{M} := \langle \Gamma_{\mathcal{M}}, Q_{\mathcal{M}}, \delta_{\mathcal{M}}, \lambda_{\mathcal{M}} \rangle$ with output symbols $\Gamma_{\mathcal{M}} = 2^{P \cup \{goal\}}$, a non-deterministic transition function $\delta_{\mathcal{M}} : Q_{\mathcal{M}} \times Q_{\mathcal{M}} \to \{0, 1\}$, and a label function $\lambda_{\mathcal{M}} : Q_{\mathcal{M}} \to \Gamma_{\mathcal{M}}$.

We define the *product automaton* as a transition system $\mathfrak{P} = \mathcal{M} \otimes \mathcal{C} := \langle Q_{\mathfrak{P}}, \delta_{\mathfrak{P}}, q_{init}^{\mathfrak{P}}, \lambda_{\mathfrak{P}} \rangle$ as follows:

$$Q_{\mathfrak{P}} := Q_{\mathcal{M}} \times Q$$
$$\delta_{\mathfrak{P}}((p,q)) := \{(p',q') \in Q_{\mathfrak{P}} | \delta(q, \lambda_{\mathcal{M}}(p) \cap \Sigma, a, q') = 1 \text{ and } \delta_{\mathcal{M}}(p,p') = 1, \text{for some } a \in A\}$$
$$q_{init}^{\mathfrak{P}} := \{(p,q_0) | p \in Q_{\mathcal{M}}\}$$
$$\lambda_{\mathfrak{P}}((p,q),(p',q')) := \{\lambda_{\mathcal{M}}(p) \cup a | a \in A \text{ and } \delta(q, \lambda_{\mathcal{M}}(p) \cap P_{\mathcal{M}}, a, q') = 1 \text{ and } \delta_{\mathcal{M}}(p,p') = 1\}$$

$\delta_{\mathfrak{P}} : Q_{\mathfrak{P}} \to 2^{Q_{\mathfrak{P}}}$ is a non-deterministic transition function, and $\lambda_{\mathfrak{P}} : Q_{\mathfrak{P}} \times Q_{\mathfrak{P}} \to 2^{P \cup P_A}$ is a label function.

The product automaton generates infinite-length trajectories in the form of $(p_0, q_0), (p_1, q_1), \ldots$ by beginning in an initial state $q_{init}^{\mathfrak{P}}$ and following the nondeterministic transition function $\delta_{\mathfrak{P}}$ thereafter. Labeled trajectories are then generated by applying the labeling function $\lambda_{\mathfrak{P}}$ to these trajectories within the product automaton, i.e. $\psi_0 \psi_1, \ldots \in (2^{P \cup P_A})^*$ where $\psi_i \in \lambda_{\mathfrak{P}}((p_i, q_i), (p_{i+1}, q_{i+1}))$. When using the product automaton to solve the model-checking problem, we check that all possible labeled trajectories generated by the product automaton belong to the language defined by the LTL specification.

# Additional Explanations to the Method

**Formal Verification vs. Empirical Evaluation.** Formal verification provides a mathematical guarantee on whether a specification is satisfied, while empirical evaluation examines the controller's behavior in practical operations.

**Definition 1.** Let $\mathcal{M}$ be the automaton-based model for the system $\mathcal{S}$. If $\mathcal{M}$ captures all the transitions $\{(\sigma, \neg\sigma) | \sigma \in P \cup P_A\}$ allowed by $\mathcal{S}$, then we say $\mathcal{M}$ captures **complete information** from $\mathcal{S}$.

From the properties of formal verification, we can derive Theorem 1.

**Theorem 1.** If $\mathcal{M}$ captures complete information from $\mathcal{S}$, then

$$\mathcal{M} \otimes \mathcal{C} \models \Phi \implies \mathbf{G}(\mathcal{C}, \mathcal{S}) \models \Phi. \tag{3}$$

*Proof.* Suppose $\mathbf{G}(\mathcal{C}, \mathcal{S}) \not\models \Phi$, then we can find at least one sequence $(\sigma_i, a_i)^N \in (2^P \times 2^{P_A})^N$ that violates $\Phi$, i.e., $(\sigma_i, a_i)^N$ is a counter-example. Hence, we get $\mathcal{M} \otimes \mathcal{C} \not\models \Phi$. By contra-positive, Theorem 1 holds. $\square$

Therefore, the formal verification results provide stronger guarantees than empirical results.

However, if the model $\mathcal{M}$ does not capture complete information from the system, then the guarantees provided by formal verification are no longer valid. Hence, we can use empirical evaluation in place of formal guarantees. From another perspective, empirical evaluation can be used to check whether the model $\mathcal{M}$ has captured complete information.

# Additional Experiment

## Demonstration

**Additional System Modeling** We present more examples of the automaton-based models encoding other scenarios in the autonomous system. The scenarios include a traffic light with an explicit left-turn signal, a two-way stop sign, and a roundabout.

**The Complete Set of LTL Specifications.** We verify the LLM's outputs through the following 15 LTL specifications:

$\Phi_1 = \square(\text{pedestrian} \to (\Diamond \text{stop}))$,
$\Phi_2 = \square(\text{opposite car} \wedge \neg\text{green left-turn light} \to \neg\text{turn left})$,
$\Phi_3 = \square(\neg\text{green traffic light} \to \neg\text{go straight})$,
$\Phi_4 = \square(\text{stop sign} \to \Diamond \text{stop})$,
$\Phi_5 = \square(\text{car from left} \vee \text{pedestrian at right} \to \neg\text{turn right})$,
$\Phi_6 = \square(\text{stop} \vee \text{go straight} \vee \text{turn left} \vee \text{turn right})$,

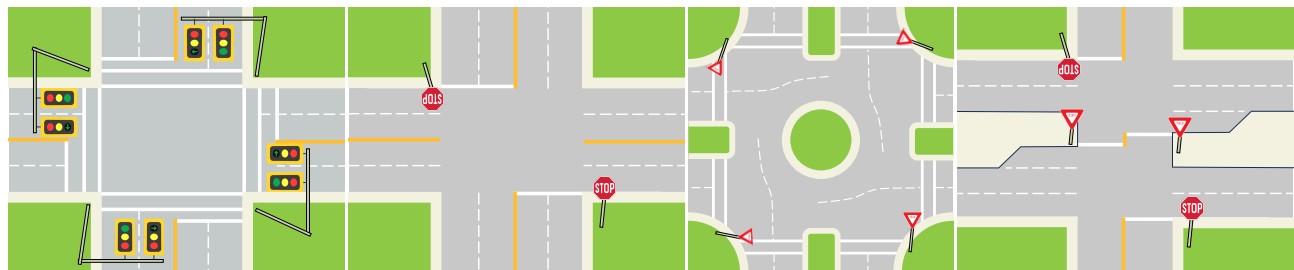

Figure 12: Illustration of different scenarios from the autonomous driving system. The figures from left to right show the scenarios for an intersection with a traffic light, a two-way stop sign, a roundabout, and an intersection with a wide median, respectively.

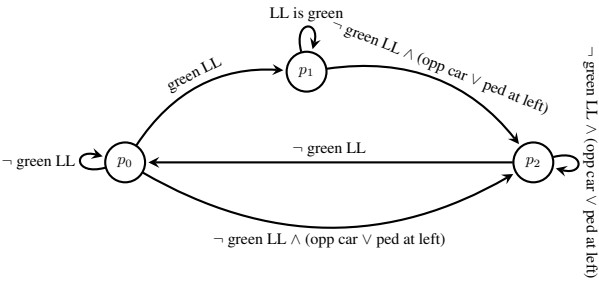

Figure 13: An automaton-based model represents a vehicle's environment dynamics at an intersection's left turn traffic signal. LL represents "Left-Turn Light," ped represents "pedestrian."

$\Phi_7 = \Diamond(\text{green traffic light} \lor \text{green left-turn light}) \to \Diamond \neg\text{stop})$,
$\Phi_8 = \Box(\neg\text{green traffic light} \to \Diamond \text{stop})$,
$\Phi_9 = \Box(\text{car from left} \to \neg(\text{turn left} \lor \text{turn right}))$,
$\Phi_{10} = \Box(\text{green traffic light} \to \Diamond \neg\text{stop})$,
$\Phi_{11} = \Box((\text{turn right} \land \neg\text{green traffic light}) \to \neg\text{car from left})$,
$\Phi_{12} = \Box((\text{turn left} \land \neg\text{green left-turn light}) \to (\neg\text{car from right} \land \neg\text{car from left} \land \neg\text{opposite car}))$,
$\Phi_{13} = \Box((\text{stop sign} \land \neg\text{car from left} \land \neg\text{car from right}) \to (\Diamond \neg\text{stop}))$,
$\Phi_{14} = \Box((\text{go straight} \to \neg\text{pedestrian in front})$,
$\Phi_{15} = \Box((\text{turn right} \land \text{stop sign}) \to \neg\text{car from left})$.

**Example on Controller Construction and Verification: Left-Turn.** We provide an example of the controllers for the task "turn left at the traffic light" and a model representing the traffic light environment. When implemented in the model, we can verify whether the controllers satisfy the provided specifications.

The responses from the language model before fine-tuning:

```
Steps for "turn left at traffic light"
1. Approach the traffic light with a left-turn light.
2. Wait for the left-turn light to turn green.
3. When the left-turn light turns green, wait for oncoming traffic to clear before turning
    left.
4. Turn left and proceed through the intersection.
```

The responses from the language model after fine-tuning:

```
Steps for "turn left at traffic light"
1. Approach the traffic light and observe the left turn light.
2. If the left turn light is not green, then stop.
3. If the left turn light is green, then turn left.
```

We present constructed controllers in Figure 16.

We implement both controllers in the model presented in Figure 13, i.e., get the product automaton of the controller and the model. Then, we can verify both controllers against the 15 specifications.

The controller obtained before fine-tuning fails specification $\Phi_{12}$, while the one after fine-tuning passes all the specifications.

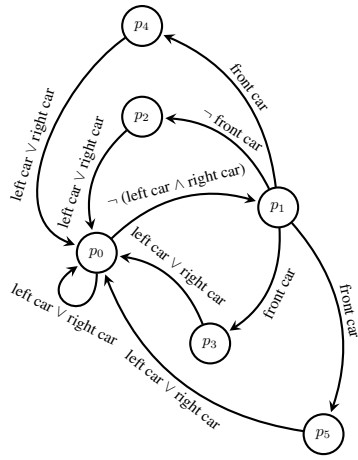

Figure 14: An automaton-based model represents a vehicle's environment dynamics at a two-way stop sign.

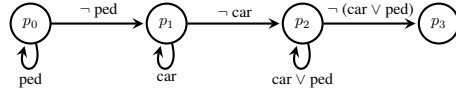

Figure 15: An automaton-based model representing the environment dynamics of a vehicle at a roundabout. The proposition "car" represents "car from left" and the proposition "ped" represents "pedestrian at left ∧ pedestrian at right."

## Verification using NuSMV

```
MODULE turn_left_before_finetune

VAR
  green_traffic_light : boolean;
  green_left_turn_light : boolean;
  opposite_car : boolean;
  car_from_left : boolean;
  car_from_right : boolean;
  pedestrian_at_left : boolean;
  pedestrian_at_right : boolean;
  side_car : boolean;
  stop_sign : boolean;
  action : {stop, turn_left, turn_right, go_straight};

ASSIGN
  init(action) := stop;

TRANS
  case
    !green_left_turn_light : next(action) = stop;
    green_left_turn_light & !opposite_car & !car_from_left & !car_from_right & !
    pedestrian_at_left & !pedestrian_at_right : next(action) = turn_left;
    opposite_car | car_from_left | car_from_right | pedestrian_at_left | pedestrian_at_right
    : next(action) = stop;
    action = turn_left : next(action) = go_straight;
    TRUE : next(action) = stop;
  esac;

MODULE turn_left_after_finetune

VAR
  green_traffic_light : boolean;
  green_left_turn_light : boolean;
```

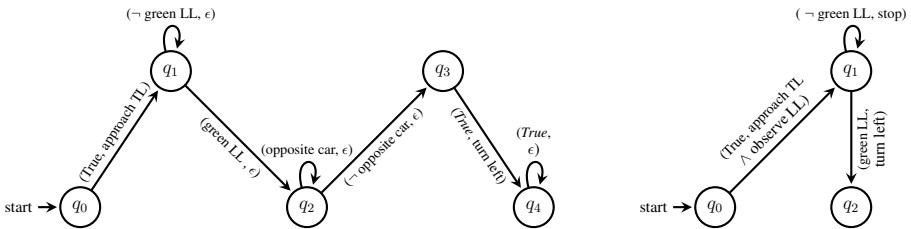

Figure 16: Automaton-based controllers for the task "turn left at the traffic light with the left-turn signal." The left and right controllers are constructed from the language model before and after fine-tuning, respectively. TL and LL represent "traffic light" and "left-turn light."

```
33   opposite_car : boolean;
34   car_from_left : boolean;
35   car_from_right : boolean;
36   pedestrian_at_left : boolean;
37   pedestrian_at_right : boolean;
38   side_car : boolean;
39   stop_sign : boolean;
40   action : {stop, turn_left, turn_right, go_straight};
41
42 ASSIGN
43   init(action) := stop;
44
45 TRANS
46   case
47     !green_left_turn_light : next(action) = stop;
48     green_left_turn_light : next(action) = turn_left;
49   esac;
50
51 LTLSPEC NAME sample_phi_1 :=
52     G( action=turn_left -> (left_turn_light in {flashing, green}) );
53
54 LTLSPEC NAME sample_phi_2 :=
55     G( F action=turn_left );
```

```
1 MODULE turn_right_before_finetune
2
3 VAR
4   green_traffic_light : boolean;
5   green_left_turn_light : boolean;
6   opposite_car : boolean;
7   car_from_left : boolean;
8   car_from_right : boolean;
9   pedestrian_at_left : boolean;
10  pedestrian_at_right : boolean;
11  side_car : boolean;
12  stop_sign : boolean;
13  action : {stop, turn_left, turn_right, go_straight};
14
15 ASSIGN
16   init(action) := stop;
17   next(action) :=
18     case
19       green_traffic_light & !car_from_right : {go_straight, turn_right};
20       green_traffic_light & car_from_right : go_straight;
21       !green_traffic_light : stop;
22     esac;
23
24 MODULE turn_right_after_finetune
25 VAR
26   green_traffic_light : boolean;
```

```
27    green_left_turn_light : boolean;
28    opposite_car : boolean;
29    car_from_left : boolean;
30    car_from_right : boolean;
31    pedestrian_at_left : boolean;
32    pedestrian_at_right : boolean;
33    side_car : boolean;
34    stop_sign : boolean;
35    action : {stop, turn_left, turn_right, go_straight};
36 ASSIGN
37    init(action) := stop;
38 TRANS
39    case
40      !car_from_left & !pedestrian_at_right : next(action) = turn_right;
41      car_from_left | pedestrian_at_right : next(action) = stop;
42    esac
43
44 LTLSPEC NAME sample_phi_1 :=
45      G( pedestrian_at_right -> ! action=turn_right );
46
47 LTLSPEC NAME sample_phi_2 :=
48      G( car_from_left -> ! action=turn_right );
```

```
1 #!NuSMV -source
2 read_model -i right_turn.smv # file name
3 go
4
5 check_ltlspec -P "phi_1" -o result1.txt
6
7 check_ltlspec -P "phi_2" -o result2.txt
8
9 quit
```

## Fine-tuning Llama-2: Implementation Details

**Llama-2 Prompts**  LLama-2 has particular implementation requirements for the prompt. Certain tokens are required which delineate system and user messages. System messages describe what role the language model should act as, while user messages describe the task at hand. We use the following prompt for Llama-2, where italicized symbols represent special input tokens, and the last sentence is a given task:

$[INST]<<SYS>>$
You are a helpful assistant. Always answer as helpfully as possible, while being safe. Your answers should be detailed.
$<</SYS>>$
Steps for "turn right at traffic light": $[/INST]$

**Fine-tuning Efficiency**  Due to hardware limitations, fine-tuning all model parameters is impractical. Instead, it is possible to fine-tune a low-rank approximation of a given matrix within the model (Hu et al. 2021). For example, instead of updating a matrix $W \in \mathcal{R}^{d \times d}$, it is more memory efficient to update two matrices $A \in \mathcal{R}^{d \times k}, B \in \mathcal{R}^{k \times d}$ with $k << d$, by holding $W$ constant and defining $\tilde{W} = W + AB$. Then, $A$ and $B$ can be updated using gradient descent with a smaller memory profile than updating $W$ itself, since the combined number of parameters in $A$ and $B$ is much less than $W$.