# OpenReview forum: "Fine-Tuning Language Models Using Formal Methods Feedback"
_AAAI.org/2024/Workshop/NuCLeaR — NuCLeaR 2024_

### Official Review · Reviewer_F4t4 · 2023-12-07
**Formal methods feedback**

**Rating:** 7
**Confidence:** 4

**Review:**

I found the idea straight forward and the work was carried out well. Sections on interfacing with simulators and formal methods are a little dry and can be cut shorter to talk more about limitations, adaptations to new domains, next steps, etc.

---

### Official Review · Reviewer_5Pf9 · 2023-12-07
**The paper talks about fine tuning using RLHF and Fine-Tuning via Automated Feedback**

**Rating:** 7
**Confidence:** 5

**Review:**

I like the idea of applying induced automata into language models fine-tuning. The experimental results especially controller construction and formulation of steps are good. The flow of the reading could have been better.

---

### Official Review · Reviewer_TC8Q · 2023-12-09

**Rating:** 8
**Confidence:** 3

**Review:**

Strengths:

1. This paper proposes direct preference optimization via automated feedback (DPO-AF), a novel method for planning that uses formal verification to create preference data and then fine-tunes the LLM-based controller with this data using DPO algorithm.

2. The authors use autonomous driving system as the environment to develop this method and results using Carla simulator shows the performance improvement

2. Overall, this is a well-rounded paper, with strong motivation, technical contribution, and evaluation section. The paper is clearly written and the contributions are well-positioned with respect to existing research.




Weaknesses:

1. Only one environment is used for experiments

---

### Decision · Program_Chairs · 2023-12-11

Accept